# Impact of Community-Based Governance Mechanisms on Transaction Intention on a Second-Hand Trading Platform

**Yuru Liu, Yan Wan * and Jun Kang**

School of Economics and Management, Beijing University of Posts and Telecommunications,
Beijing 100876, China
* Correspondence: wanyan@bupt.edu.cn

**Abstract:** Second-hand trading platforms are helpful to the recycling of resources. It is important to accelerate the construction of second-hand trading platforms and improve people's willingness to buy second-hand goods. However, due to the uncertainty of second-hand goods, it is difficult to establish the trust between users and complete second-hand transactions. Nowadays, more and more platforms use community-based governance mechanisms to promote relationships between users. Taking the second-hand trading platform Xianyu as an example, this study explores the influence of three specific community-based mechanisms (interest group, feedback mechanism and dispute resolution mechanism) on trust and transaction intention from three dimensions of relational governance. This study compares the different effect between consumers and prosumers. Based on 721 valid questionnaires, a structural equation model was used to analyze the data. The results show that interest group, feedback mechanism and dispute resolution mechanism all have significant positive effects on trust in sellers and platforms. In addition, the impact of a dispute resolution mechanism on trust in sellers and platforms is higher for prosumers than for consumers. This study extends the previous research on community-based governance, contributes to the design of second-hand trading platforms and promotes more users to participate in recycling economy.

**Keywords:** second-hand trading platform; platform governance; community-based governance mechanisms; trust; prosumer

## 1. Introduction

The term second-hand trading platform refers to a platform where individuals provide or buy second-hand resources with one another. With the improvement of people's economic awareness and environmental awareness, more and more people are involved in second-hand trading platforms [1,2]. According to QYResearch, the commodity turnover of global second-hand trading platforms reached USD 581.5 billion in 2020, with a compound annual growth rate of 19.66%. Among them, the Chinese market was USD 192.9 billion, accounting for 33% of the total, and the American market was USD 139.6 billion, accounting for 24%. Although the second-hand market is developing rapidly, the uncertainty of sellers and products on second-hand trading platforms is stronger than in traditional e-commerce platforms [3,4]. Compared with traditional e-commerce platforms, it is more critical to enhance the governance ability of second-hand trading platforms and enhance users' trading willingness.

Interaction with social members can mediate the association from social media to continuance intention [5], and the sense of virtual community has significant positive effects on consumer–brand relationships [6]. So, current second-hand trading platforms integrate virtual community into e-commerce websites and assist traditional regulatory strategies (such as supervision, escrow, etc.) by establishing communities [7]. The virtual community on e-commerce platforms can influence customer loyalty and participation through the formation of social support and community identity, trust in the community and other

community factors [8,9]. At the same time, different levels of community participation also lead to different purchasing behaviors of users. The higher the level of community participation, the easier it is for customers to consume more products/services [10]. Users' opportunistic behaviors also can be reduced [11]. In addition, there is an interaction between community and e-commerce. Virtual communities and e-commerce can jointly influence customer engagement behavior through trust and perceived risk [12]; however, the rich social activities in virtual communities will reduce the positive impact of the e-commerce service [7].

Previous studies took the community as a whole and explored the role of social interaction in the community on the e-commerce platform. However, it is necessary to establish and maintain the relationship between users through many specific governance mechanisms (such as interest groups, feedback evaluation, dispute resolution, etc.) from different dimensions. From the perspective of governance, the current research is not clear on how different dimensions of community-based governance on the platform affect users. In addition, a few studies have discussed the differences between users with different levels of community participation [10]; however, consumers participate in second-hand transactions in various forms, including traditional consumers (who only participate in purchases) and prosumers (who participate in purchases and sales). It is not clear how the impact of community-based governance on these two groups of consumers differs. So, this study is guided by the following two research questions: How do different dimensions of community-based governance affect consumers' transaction intentions? What is the difference between the impact of community-based governance on consumers and prosumers?

Community can build relationships among market participants and improve platform performance, which can be considered as a relational governance [13]. In order to better understand how community-based governance on the platform affect consumers' transaction intentions, this study explores community-based governance by building a model based on three dimensions of relational governance (relational norms, conflict resolution and mutual dependence). At the same time, this study also uses multi-group analysis to explore the differences in the impact of community-based governance between consumers and consumers on a second-hand trading platform.

This study regards community building as a new form of governance, explores the impact of community-based governance, clarifies the boundary role of community-based governance, contributes to the current literature in the field of platform governance and online community and provides suggestions for platform managers on how to build communities and make more effective use of community-based governance.

## 2. Theoretical Background

### 2.1. Online Community

Online community in e-commerce is a cyberspace where buyers and sellers gather together with certain rules and norms [14]. The virtual community enriches the interactive forms of users and promotes mutual communication so as to smooth the transaction [15]. These interactions allow users to build relationships between themselves and foster a sense of commitment to the community [16]. In online communities, members exchange information, share experiences and achieve specific goals through collaboration [17].

As a kind of social network space, community is built in many forms. Community building is inseparable from network-based communication technology. Some platforms offer chat rooms or bulletin boards [18]. Computer communication tools (instant messaging, message boxes) can help buyers and sellers establish swift guanxi through interaction and social presence [19]. Online forums are social environments that promote individual social interaction and are one of the structures of community building. Members of online forums participate in group activities together and support other members through their social interaction [20]. Feedback and comments are also part of community building. Individuals can post their product reviews and merchant reviews [21]. They also provide advice and

share experiences about products and merchants on these platforms, providing a source of online social support [22]. Based on the different tightnesses of the above community infrastructure, coupling relationships among community members is different [11].

Online communities can bring information value and social value [23]. The community can form social capital to maintain the membership relationship in the community [24]. The social environment increases customers' purchase intentions through interaction [25–27]. The interaction between users can enhance the intimacy and familiarity between users to enhance the transaction intention of users [28]. The sharing awareness, sharing rituals, sense of responsibility and organizational commitment formed by users in the community can effectively improve users' trust and loyalty to the community [16,29]. Through ongoing community involvement, members spontaneously contribute to the community, resulting in commitment and shared values [30]. Social support and community factors in the community (community drivenness, community identification, community trust) affect customer loyalty and customer engagement [8]. Community in e-commerce reduces users' opportunistic behavior through social interaction and information sharing among virtual community members [11].

### 2.2. Governance Mechanism

Governance is a means by which to infuse order and thereby mitigate conflict and realize mutual gain [31]. We adopt the definition of Ceccagnoli and believe that the platform governance mechanism refers to specific services and policies provided by the platform owners for platform users [32], through which market conditions can be improved, user participation can be stimulated and friction can be solved [33]. Transaction cost economics (TCE) is regarded as the main theoretical perspective to the governance mechanism of the relationship between buyers and sellers. Reducing transaction costs by means of various control mechanisms is fundamental to the transaction relationship [34]. Institutional constraints are needed to reduce uncertainty in the transaction process [35].

TCE proposes two main governance mechanisms (namely, formal contractual mechanisms and relational mechanisms) to regulate the relationship between buyers and sellers, both of which are developed by the institutions that protect transactions [36]. Formal governance mechanism focuses on the formulation of norms, constrains the behaviors of buyers and sellers by taking contracts as the medium and formulates a series of reward and punishment measures [37]. The formal contract mechanism cannot explain all possible contingencies in transactions [38]. The relational governance mechanism is used to complement the formal contract mechanism. The relational governance mechanism encourages interactive behaviors, mainly through the development of social relations and sharing norms [38]. Social control stimulates the enthusiasm to perform contracts due to the mutual identification between partners and complements formal control [39]. These mechanisms cover flexible issues in trading and encompass dimensions of relationship norms, conflict resolution and mutual dependence [40].

On Xianyu, as a second-hand trading platform, the second-hand goods are more or less defective. It is difficult for formal governance to bind the quality of each commodity to a fixed standard, and a more resilient governance mechanism is needed on the platform. Therefore, Xianyu can reduce opportunistic behavior on the platform through the integration of community functions and the establishment of social relations between buyers and sellers. Community-based governance mechanism is a mechanism to build embedded relationships on the platform, which is essentially a form of relational governance. Therefore, this study explores community-based governance mechanisms from the perspective of relational governance.

### 2.3. Trust in Sellers and Trust in Platform

On the Internet, due to its unique characteristics of spatiotemporal separation, there is a serious problem of information asymmetry in the transaction process that will lead to a variety of uncertain factors. Therefore, the establishment of trust is particularly

important [41]. Trust is where people and organizations can rely on each other and trust each other regardless of the uncertainty of the future [42].

A platform is a tool to connect buyers and sellers. To reach a deal, buyers must first choose a trusted platform and then choose a trusted seller. For consumers, the objects of trust mainly include platforms and sellers [41,43,44]; many scholars have conducted research on these two types of trust. Consumers' trust in the platform is mainly based on the establishment of cognitive factors, which have an impact through the institutional guarantee provided by the platform. Firstly, the platform can formulate security measures to enhance consumers' trust, such as authentication, encryption, etc. [45]. Secondly, the quality of the platform [45,46], including website quality and service quality, is a key factor affecting trust. Finally, the external reputation of the platform will also have an impact on consumer trust [47]. Consumer trust in the seller is a kind of interpersonal trust that is generated not only through the seller's personal trust, including personal profile, evaluation information and background information [48–50], but also through emotional factors such as the similarity and interactive communication between the two sides [51,52].

*2.4. Prosumer*

Participants in the platform economy are divided into suppliers and customers [53,54]. This division implies a basic logic: the roles of the suppliers and the customer are relatively single and fixed and cannot be easily switched between the two. However, in the current platform, users often play both supplier and customer roles. Prosumers generally refer to those individuals who can not only create value by participating in production activities but also enjoy value through consumption activities [55]. Specifically, based on the original consumption activities, they can create value through leasing, borrowing, sharing, exchange, transaction and other forms of production activities so as to realize the integration of production and consumption [56]. It includes both information content production activities on social e-commerce platforms [57] and various service activities by individual prosumers on digital platforms using personal assets, such as short-term rental and second-hand goods trading [58]. Digital technology enables a high level of autonomy, and the roles of production and consumption are easily changed. They can participate in consumption activities as consumers at one point in time and participate in production service activities as producers at another point in time [59].

The prosumers on the platform have the same status and power as consumers and do not practice the logic of "Customer First" in traditional service. For example, some landlords of short-term rental platforms can even refuse consumers' demands [60]. The interactive relationship between prosumers and consumers is different from the relationship between traditional producers and customers, which tends to be the same exchange relationship between the two sides. Therefore, the harmonious relationship in service interaction is important [61,62]. As a relationship embedding mechanism, it is meaningful to explore the influence of community-based governance mechanisms on both sides of this new relationship. This study will explore the impact of community-based governance mechanisms on prosumers and consumers and compare the differences between them.

## 3. Research Models and Hypothesis

This study proposes a model to explain the impact of community-based governance mechanisms on second-hand trading platforms. As shown in Figure 1, the model reflects that two types of trust can be influenced on the platform, thus ultimately affecting the transaction intention. At the same time, the identity of users has a moderating effect on the relationship between the dispute resolution mechanism and two types of trust. In the latter part of this paper, the research model is defined in detail and corresponding hypotheses are proposed.

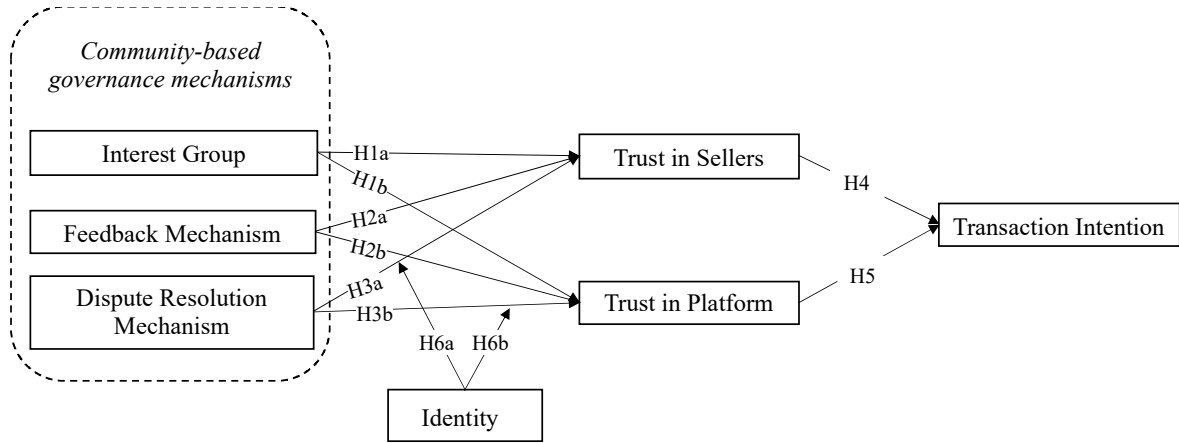

**Figure 1.** Research model.

*3.1. Community-Based Governance Mechanisms and Trust*

A community-based governance mechanism represents a mechanism that builds an embedded relationship on the platform, which is essentially a relationship governance. Goo, Kishore and Rao [40] pointed out that relationship governance refers to the role of obligation, commitment and expectation implementation through trust and social identity, including three dimensions of relationship norms (information exchange, unity and flexibility), conflict resolution and mutual dependence. According to these three dimensions, this paper selected three innovative governance mechanisms on second-hand trading platforms.

*3.2. Interest Group*

Among the three dimensions of relational mechanisms, normative relational mechanisms foster mutually accepted and expected patterns of behavior that are directed towards collective goals [40,63]. Interest group mechanism gathers users with common interest. Buyers and sellers on the platform can join or form groups freely according to their interest. Interest groups build solidarity, expect both sides to form a common pattern of behavior that considers each other's interests, and enable users to contribute to the community.

Interest groups are social environments that promote individual social interaction. Based on their own interests, users participate in different groups and support other members through their social interaction and communication [20]. For the sustainable operation of the community, members of the community can formulate common norms to manage the behavior of members in the community. Members in the same interest group who discuss common topics are similar. According to the studies [64,65], the perceived similarity between members is the antecedent of trust in other members. Similarly, if buyers and sellers are in the same community and have the same hobbies, it is easier to build trust. The intimacy and familiarity formed by the interaction between members are also important antecedents of mutual trust among members [64]. In addition, group members share information, experiences and other content with each other, which can form a social reciprocal relationship [66]. Social relationships can strengthen the connection between people and enhance individuals' trust in others [67].

Users participate in interaction and deepen their contact with others, which can effectively improve and enhance the sense of social presence and social support on the platform [68,69]. The social support and relationship quality on the platform can have a positive impact on users and improve their trust in platform [22]. In addition, the discussion of online products can reduce the uncertainty in the product and increase the trust in the platform [70].

Therefore, we propose the following hypotheses:

**H1a.** *Interest groups can positively affect buyers' trust in sellers.*

**H1b.** *Interest groups can positively affect buyers' trust in platforms.*

### 3.3. Feedback Mechanism

Mutual dependence refers to making both parties realize they should depend on each other in order to obtain the interests of the program [40]. For example, a feedback system is a kind of interdependent mechanism that helps to form a common cognition, and their success depends on the good opinion of the other side. This mechanism provides a sense of dependency, thereby motivating participants to place high priority on partnerships [71].

A feedback mechanism is a mechanism for mutual evaluation between buyers and sellers on the platform. Online feedback communities are widely used on e-commerce platforms [72], such as the feedback forum on eBay where users can post their comments on buyers and sellers and their experience of buying and selling goods. In the feedback mechanism, buyers check the sellers' comments and previous transaction information before the transaction. After the transaction, the buyer and the seller evaluate and score each other [73]. For users on the platform, it is easy to understand the information about goods and sellers through user feedback [19]. The feedback mechanism on the platform can distinguish different sellers and build reputation for sellers at the lowest cost [21,48]. The feedback mechanism collects information about the seller's past transaction behavior, which provides a reference for the buyer to establish trust in the seller.

For users on the platform, it is easy to get the information about goods and sellers through user feedback. The feedback mechanism can effectively promote information transmission and reduce information asymmetry in the shopping process [19]. The feedback mechanism is also a market signaling mechanism that can reduce product uncertainty [3,74]. Those signs on the platform lead to a subconscious feeling within the users of more security and, in turn, more trust [47].

Therefore, we propose the following hypotheses:

**H2a.** *Feedback mechanisms can positively affect buyers' trust in sellers.*

**H2b.** *Feedback mechanisms can positively affect buyers' trust in platforms.*

### 3.4. Dispute Resolution Mechanism

A dispute resolution mechanism is one that helps parties reach satisfactory solutions by reaching a consensus in a dispute [40,75]. Due to the complexity of second-hand goods trading, it is difficult for consumers to judge the quality of products before transaction and, as consumers are not allowed to return or exchange goods after transaction, it is prone to disputes. The dispute resolution mechanism is a mechanism for the platform to help resolve disputes between buyers and sellers [72]. Therefore, a high-quality dispute resolution mechanism can help the buyers and sellers to coordinate their relationship, which is particularly important for the smooth transaction of second-hand goods.

Online dispute resolution mechanisms can resolve disputes between buyers and sellers on the platform in a convenient and efficient way [76]. The judges of disputes on the platform are third parties with irrelevant interests, who have rich shopping experience and different background knowledge and can make fair judgments on complex disputes [77]. The high-quality dispute resolution mechanism on the platform means that the community can effectively deal with disputes and minimize the risk caused by the opportunistic behavior of sellers [72]. For disputes caused by sellers' opportunism, a fair solution mechanism can punish sellers and thus limit sellers' speculative behavior. Buyers are more likely to believe that sellers on the platform will not exhibit opportunistic behaviors, and their trust in sellers will be easily established [78].

When there is a problem, the dispute resolution mechanism can protect the interests of users and reduce the risk of buyers in the transaction, which is also a kind of security mechanism [72]. Perception of security protection enhances buyers' confidence in the transaction process and increases the buyer's trust in the platform [47].

Therefore, we propose the following hypotheses:

**H3a.** *Dispute resolution mechanisms can positively affect buyers' trust in sellers.*

**H3b.** *Dispute resolution mechanisms can positively affect buyers' trust in platforms.*

*3.5. Trust and Transaction Intention*

Trust is a crucial factor on the shopping platform. For buyers, the seller is the object of the transaction and an important part of the transaction process. Trust is the belief in a person's integrity, benevolence and ability [79].

When the buyer trusts the seller, it means that the buyer believes that the seller's behavior is in line with expectations. At the same time, buyers' trust in sellers can effectively reduce users' perceived risk and alleviate information asymmetry between buyer and seller [48]. Trust can enhance users' satisfaction, and the improvement of buyers' satisfaction contributes to users' transaction intention [80]. When the buyer trusts the platform, it indicates that the buyer believes that the transaction on the platform meets the expectations [47], which represents the buyer's trust in the information and services on the platform [81], and is naturally willing to trade on the platform. Current research has confirmed that the trust of buyers on e-commerce platforms can reduce their uncertainty, reduce their concerns about security and privacy and increase the perceived usefulness of the platform [82].

Therefore, we propose the following hypotheses:

**H4.** *Trust in sellers can positively affect the buyers' transaction intention.*

**H5.** *Trust in platforms can positively affect buyers' transaction intention.*

*3.6. The Moderating Effect of User Roles*

On second-hand trading platforms, buyers and sellers participate in transactions for different purposes. Buyers pursue price advantages and uniqueness of goods [83], whereas sellers mainly seek economic income and social value [84] The mechanism of trust establishment between buyers and sellers on the platform is also different [43]. This paper mainly discusses the differences between consumers (the people who only buy goods) and prosumers (the people who both buy and sell goods). In other words, how does the governance mechanism affect buyers who also act as a seller.

Studies have shown that technology can empower prosumers with higher autonomy and allows them to carry out production activities in a wider market [85]. Digital technology empowerment covers both process and result [86]; in process, digital technology enables individuals (such as prosumers) to participate more in the activities on the platform, know more about the factors in the market environment and enhance their understanding on the decision-making process. According to the empowerment theory [86], individual prosumers can better understand governance mechanisms. Familiarity can enhance perceived ease of use of the governance mechanism and reduce the complexity of the decisions, thus can increase the effect of the governance mechanism [87,88]. In addition, if prosumers have more sense of autonomy and control then they are more satisfied with the participation of the governance mechanism on the platform [89], thus enhancing the effect of the governance mechanism on trust in sellers.

However, we should consider the three specific forms of the community-based governance mechanisms. In interest groups, both buyers and sellers can freely participate in group activities and discussions. For feedback mechanisms, both the buyer and the seller can evaluate each other after the purchase. For both governance mechanisms, the buyer and the seller are in an equal position. Although prosumers have the dual roles of buyer and seller, their dual role cannot enhance the understanding of these mechanisms. However, in dispute resolution mechanisms, buyers and sellers participate in the process as plaintiffs and defendants, respectively. In this case, individual prosumers can better understand the governance mechanism than ordinary consumers, enhancing the role of trust in sellers and platforms.

Therefore, we propose the following hypotheses:

**H6a.** *The impact of dispute resolution mechanisms on trust in sellers is higher for prosumers than for consumers.*

**H6b.** *The impact of dispute resolution mechanisms on trust in platforms is higher for prosumers than for consumers.*

**4. Research Methodology**

*4.1. Method and Data Collection*

From 2 December 2021 to 30 February 2022 a total of 800 questionnaires were distributed to users who has traded on Xianyu on Tencent's questionnaire platform. To maximize the response rate, we provided each respondent with a cash bonus. After excluding the invalid questionnaires that were illogical, 721 valid questionnaires were received. The valid samples met the suggestion that the sample size should be at least 10 times the test items [90]. Table 1 summarizes the descriptive statistical analysis of respondents' demographics. The collected samples are relatively balanced in gender, mostly under the age of 25, most of them have bachelor's degree and the frequency of platform use is once a month or once a week.

**Table 1.** Demographics of survey respondents (*N* = 721).

| Demographic Profile | Categories | Full Sample | |
| --- | --- | --- | --- |
| | | **Frequency** | **Percent (%)** |
| Gender | Male | 314 | 43.6 |
| | Female | 407 | 56.4 |
| | Below 18 | 30 | 4.2 |
| | 19–25 | 517 | 71.7 |
| Age (in years) | 26–35 | 132 | 18.3 |
| | 36–45 | 16 | 2.2 |
| | Above 46 | 6 | 0.8 |
| | High school or below | 206 | 28.6 |
| Education | College | 468 | 64.9 |
| | Graduate or above | 47 | 6.5 |
| | Seldom | 128 | 17.8 |
| Frequency of platform | Once a month | 238 | 33.0 |
| | Once a week | 305 | 42.3 |
| | Once a day | 50 | 6.9 |

*4.2. Measurement*

To verify the validity of the hypothesis, this study conducted empirical analysis through survey data. The questionnaire design is divided into two parts which can be found in Appendix A. The first part collects basic information, including gender, age, education background and frequency of platform use. In the second part, seven variables are established according to the research hypotheses proposed above and specific questions are adjusted according to the second-hand transaction. The constructs involved are shown in Table 2, which are adapted from the existing relevant literature using 7-point Likert scale.

**Table 2.** Constructs and associated items.

| Constructs | Measurement Item | Factor Loading | $\alpha$ | CR | AVE |
|---|---|---|---|---|---|
| Interest group [91] | Users in interest group have similar interests to me. | 0.940 | | | |
| | Users in interest group share similar values to me. | 0.938 | 0.934 | 0.959 | 0.885 |
| | Users in interest group are very close to me. | 0.945 | | | |
| Dispute resolution mechanism [78] | The mechanism can protect me if the sellers try to cheat me. | 0.890 | | | |
| | The mechanism can guarantee my interest if the seller tries to provide a low-quality product/service. | 0.904 | 0.925 | 0.947 | 0.816 |
| | The mechanism has been effective in protecting my interests. | 0.916 | | | |
| | The mechanism can guarantee me a refund. | 0.904 | | | |
| Feedback mechanism [48] | The mechanism provides accurate information about a sellers' reputation. | 0.912 | | | |
| | The mechanism has access to a wealth of useful information about the sellers' transaction history. | 0.920 | 0.907 | 0.941 | 0.842 |
| | The mechanism would help me evaluate the sellers. | 0.922 | | | |
| Trust in Platform [7] | I think that Xianyu is reliable. | 0.904 | | | |
| | I think that Xianyu will keep its promise. | 0.922 | 0.908 | 0.936 | 0.785 |
| | Xianyu is a trustworthy channel for me to transact. | 0.893 | | | |
| | The service offered by Xianyu meets my expectation. | 0.822 | | | |
| Trust in Sellers [48] | Sellers in Xianyu are in general trustworthy. | 0.945 | | | |
| | Sellers in Xianyu are in general reliable. | 0.940 | 0.935 | 0.949 | 0.885 |
| | Sellers in Xianyu are in general honest. | 0.938 | | | |
| Transaction Intention [48] | I would consider transacting on Xianyu. | 0.928 | | | |
| | It is likely that I will transact on Xianyu in the near future. | 0.923 | 0.912 | 0.947 | 0.850 |
| | Given the opportunity, I intend to transact on Xianyu. | 0.915 | | | |

Note: $\alpha$ = Cronbach's alpha; CR = composite reliability; AVE = average variance extracted.

## 5. Data Analysis

### 5.1. Measurement Model Evaluation

The structural equation model can study the relationship between multiple latent variables at the same time [92]. Therefore, the structural equation model was selected in this study. In this study, the reliability and validity test of the scale is carried out using SPSS 25.0 to confirm whether the questionnaire items can reflect our purpose and accurately measure the subjective feelings of the respondents. If the data is valid, Amos 20.0 is used to verify the research hypotheses. Then, the multiple group analysis is used to compare the differences between different product types.

As shown in Table 2, the Cronbach's alpha of all constructs is greater than 0.9, demonstrating the scale is reliable [93]. The average extraction variance (AVE) of constructs is greater than 0.7 and the factor load of items is greater than 0.8, indicating that the convergence validity of the scale has also been verified [94]. Finally, as shown in Table 3, the square root of AVE of all constructs is greater than the corresponding correlation, indicating good discriminant validity [93].

**Table 3.** Discriminant validity of constructs.

| Constructs | 1 | 2 | 3 | 4 | 5 | 6 |
|---|---|---|---|---|---|---|
| (1) Interest group | 0.941 | | | | | |
| (2) Feedback mechanism | 0.441 | 0.918 | | | | |
| (3) Dispute resolution mechanism | 0.438 | 0.640 | 0.904 | | | |
| (4) Trust in platform | 0.426 | 0.543 | 0.589 | 0.886 | | |
| (5) Trust in sellers | 0.528 | 0.560 | 0.613 | 0.644 | 0.941 | |
| (6) Transaction intention | 0.354 | 0.497 | 0.455 | 0.517 | 0.552 | 0.922 |

Notes: The figures under the diagonal are the correlations between the variables. Diagonal elements are square roots of average variance extracted.

### 5.2. Coefficient Significance Test

This study uses Amos 20.0 to verify the research hypotheses. All the analysis results are shown in Figure 2, which shows that the community-based governance mechanisms explain 55.1% of the trust in sellers, trust in sellers explains 45.7% of the trust in the platform and trust in platform and trust in sellers explains 38.3% of the transaction intention, indicating that the model has a certain prediction ability.

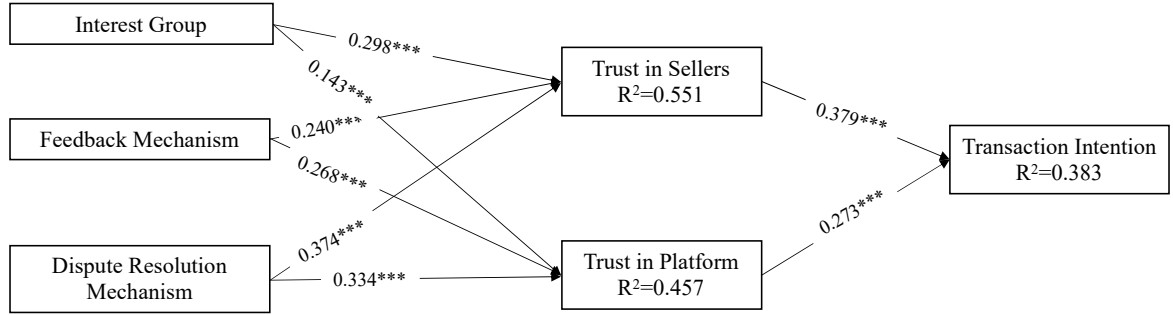

**Figure 2.** Full Sample Analysis Results. Note: *** represents at 0.001 level (*p* < 0.001).

The obtained model path coefficients and hypothesis test results are presented in Table 4. Interest group ($\beta = 0.298$, $t = 8.450$), feedback mechanism ($\beta = 0.240$, $t = 4.642$) and dispute resolution mechanism ($\beta = 0.374$, $t = 8.468$) all have a significant positive impact on trust in sellers; H1a, H2a and H3a are supported. The impact of interest group on trust in platforms ($\beta = 0.143$, $t = 4.093$), feedback mechanism ($\beta = 0.268$, $t = 5.199$) and dispute resolution mechanism ($\beta = 0.334$, $t = 7.532$) have a significant positive impact on trust in platforms; H1b, H2b and H3b are supported. Trust in sellers ($\beta = 0.379$, $t = 9.300$) and trust in platform ($\beta = 0.273$, $t = 6.109$) positively affected transaction intention, thus supporting H4 and H5.

**Table 4.** Structural model results.

| Hypotheses | Coefficient | T-Statistics | Result |
|---|---|---|---|
| H1a: Interest group → Trust in sellers | 0.298 | 8.450 | Accepted |
| H1b: Interest group → Trust in platforms | 0.143 | 4.093 | Accepted |
| H2a: Feedback mechanism → Trust in sellers | 0.240 | 4.642 | Accepted |
| H2b: Feedback mechanism → Trust in platforms | 0.268 | 5.199 | Accepted |
| H3a: Dispute resolution mechanism → Trust in sellers | 0.374 | 8.468 | Accepted |
| H3b: Dispute resolution mechanism → Trust in platforms | 0.334 | 7.532 | Accepted |
| H4: Trust in sellers → Transaction intention | 0.379 | 9.300 | Accepted |
| H5: Trust in platform → Transaction intention | 0.273 | 6.109 | Accepted |

### 5.3. Multiple Group Analysis

In order to distinguish the impact of a dispute resolution mechanism on different users, we divide the total sample into two parts: prosumers (*N* = 464) and consumers (*N* = 257). This study uses the multi-group analysis (MGA) function of Amos to compare the differences of path coefficients among multiple groups [95].

The data in Table 5 shows the path coefficient and the significance level of the structural model constructed between two group samples. There is difference between the two groups of samples. The path coefficient between dispute resolution mechanism and trust in sellers is significantly larger in the prosumers sample than in the consumers sample ($\beta_{consumers} = 0.248$, $\beta_{prosumers} = 0.456$, T = 2.305). The path coefficient between dispute resolution mechanism and trust in platforms is significantly larger in the prosumers sample than in the consumers sample ($\beta_{consumers} = 0.274$, $\beta_{prosumers} = 0.384$, T = 1.672).

**Table 5.** Path coefficient comparison between consumers and prosumers.

| Path | Coefficient | | Difference | T-Value |
|---|---|---|---|---|
| | **Consumers** | **Prosumers** | | |
| Dispute resolution mechanism → Trust in sellers | 0.248 *** | 0.456 *** | −0.208 ** | −2.305 |
| Dispute resolution mechanism → Trust in platforms | 0.274 *** | 0.384 *** | −0.110 * | −1.672 |

Note: * represents at 0.05 level ($p < 0.05$); ** represents at 0.01 level ($p < 0.01$), *** represents at 0.001 level ($p < 0.001$).

## 6. Discussion and Conclusions

### 6.1. Discussion

Drawing on the literature of online community and platform governance, our research theoretically develops a model and empirically tests the model. The results show that community-based governance plays an important role in the establishment of trust on the platform. This study contributes to the literature in the field of platform governance and online communities.

Firstly, we did not study community-based governance from a social perspective like previous studies [7,23,28]. Based on the meaning of relationship governance [40], we divided community governance into three dimensions: relationship norms, conflict resolution and mutual dependence, and discuss the impact of three specific mechanisms on trust in second-hand trading platforms. This study distinguishes between two types of buyer trust on the platform. Previous studies have shown the difference between the two types of trust: trust in platforms is based on institutional guarantees and trust in sellers is based on interpersonal interaction [43,44]. Through the research, we find that all the community-based governance mechanisms can improve transaction intention through two types of trust, in which the interest group has a higher impact on seller trust than platform trust and feedback mechanism and dispute resolution mechanism have little difference on the two types of trust. This reflects that community-based governance mechanisms can promote the enhancement of transaction intention on the platform from the two aspects of institution construction and interpersonal enhancement, but the impact of specific governance mechanisms is different. Compared with previous studies, this study confirms the role of the community in enhancing platform governance capabilities from three dimensions of relationship norms, conflict resolution and interdependence.

In addition, this research explores the impact of community-based governance mechanisms on different types of buyers. On second-hand trading platforms there is a new type of buyer, the prosumer, who participates in production activities and consumption activities. Previous studies have examined the participation motivation of prosumers [96] and the value that prosumers can provide to the platform [97] but have not considered the impact of platform governance on new production relationships. This study compares the differences between consumers and prosumers through multi-group analyses. The results show that the dispute resolution mechanism has a different impact on the two, and the dispute resolution mechanism has a significantly higher impact on consumers' trust in sellers and platforms than prosumers. This shows that the change of user identity will affect the results of community-based governance. When users participate in platform activities with multiple identities, some community-based governance will have a greater impact on users.

### 6.2. Practical Implications

The results of this study provide practical guidance for second-hand trading platform, showing them how to use community-based governance mechanisms to build trust. This is conducive to the construction of second-hand trading platforms and improving the willingness of users to buy second-hand goods, which is particularly important for environmental protection and long-term social transformation. In addition, although this research is based on a second-hand trading platform, it has significance for the establishment of communities on traditional e-commerce platforms or sharing platforms. This study is helpful

for platform enterprises to improve their governance mechanisms. Platform enterprises can build communities from three dimensions: relationship norms, conflict resolution and interdependence.

At the same time, there are many users with new identities on the on the second-hand trading platform and sharing platform who not only provide production services but also participate in consumption activities [55]. The platform needs to pay attention to the influence of the governance mechanisms on users with different identities. This paper proves that the influence of a dispute resolution mechanism on prosumers is higher than that on consumers. Therefore, the platform can formulate incentive strategies to promote the role transformation of users with a single identity and encourage buyers to participate in sales activities, which can promote users' understanding of the platform governance mechanism.

*6.3. Limitations*

This paper has made some conclusions but there are still some research limitations to be further explored. First of all, the samples in this study are from China. It is unclear whether these conclusions can be generalized to any users. In the future, samples from all over the world can be collected on this issue to facilitate the generalization of conclusions. In addition, this study mainly explores the buyer's trust in sellers, but the relationship between users in transaction is unexplored. In the future, we can consider studying the impact of governance mechanisms on sellers' trust. Last but not least, this study discusses the influence of different community-based governance mechanisms on trust but does not consider the complementary or substitutive effect among community-based governance mechanisms. Further research can be conducted that how platform enterprises coordinate various governance mechanisms.

**Author Contributions:** Conceptualization, Y.L., Y.W. and J.K.; Funding acquisition, Y.W. and J.K.; Methodology, Y.L.; Supervision, Y.W. and J.K.; Writing—original draft, Y.L. and Y.W.; Writing—review and editing, Y.L., Y.W. and J.K. All authors have read and agreed to the published version of the manuscript.

**Funding:** This research was supported by the National Natural Science Foundation of China (71874018) from Y.W. and National Natural Science Foundation of China (71772059) from J.K.

**Institutional Review Board Statement:** Not applicable.

**Informed Consent Statement:** Not applicable.

**Data Availability Statement:** Not applicable.

**Conflicts of Interest:** The authors declare no conflict of interest.

**Appendix A. Questionnaire of Transaction Intention of Buyers on Second-Hand Trading Platform**

Dear Sir/Madam, the purpose of this survey is to study your attitudes and opinions of Xianyu platform. The results of the survey will be kept strictly confidential and used for academic research only. Please make sure that you have used the Xianyu platform before you reply. We sincerely thank you for your cooperation here.

1. On the Xianyu platform, I usually [Single choice]
   - ○ Buying second-hand goods
   - ○ Selling second-hand goods
   - ○ Both, with similar frequencies

2. How often do I use the Xianyu platform? [Single choice]
   - ○ Not Often
   - ○ Several times a month
   - ○ Several times a week
   - ○ Several times a day

3. The items I usually trade on the Xianyu platform are [Single choice]
   - ○ Clothes, shoes and hats
   - ○ Electronics
   - ○ Books
   - ○ Electronic material
   - ○ Cosmetics
   - ○ Peripheral product
   - ○ Other ____

4. Your Gender [Single choice]
   - ○ Male
   - ○ Female

5. Your Age [Single choice]
   - ○ Age 18 and younger
   - ○ Age 19–25
   - ○ Age 26–35
   - ○ Age 36–45
   - ○ Age 46 and older

6. Your Education Level [Single choice]
   - ○ High school degree or less
   - ○ Undergraduate degree
   - ○ Master degreee
   - ○ Doctor degree

7. On the Xianyu platform, I think [Single choice]
   - ○ Choosing second-handgoods for economic reasons
   - ○ Exchanging goods with new friends who share common interests
   - ○ Both

8. Regarding the interest group on Xianyu, I think [Scale question]

   Users in interest group have similar interests to me.
   Strongly disagree ○1 ○2 ○3 ○4 ○5 ○6 ○7 Strongly agree
   Users in interest group share similar values to me.
   Strongly disagree ○1 ○2 ○3 ○4 ○5 ○6 ○7 Strongly agree
   Users in interest group are very close to me.
   Strongly disagree ○1 ○2 ○3 ○4 ○5 ○6 ○7 Strongly agree

9. Online dispute resolution is a mechanism for Xianyu to deal with user disputes and complaints. In my opinion, [Scale question]

   The mechanism can protect my if the sellers try to cheat me.
   Strongly disagree ○1 ○2 ○3 ○4 ○5 ○6 ○7 Strongly agree
   The mechanism can guarantee my interest if the seller tries to provide a low quality product/service.
   Strongly disagree ○1 ○2 ○3 ○4 ○5 ○6 ○7 Strongly agree
   This is a test question. Please choose number two.
   Strongly disagree ○1 ○2 ○3 ○4 ○5 ○6 ○7 Strongly agree
   The mechanism has been effective in protecting my interests.
   Strongly disagree ○1 ○2 ○3 ○4 ○5 ○6 ○7 Strongly agree
   The mechanism can guarantee me a refund.
   Strongly disagree ○1 ○2 ○3 ○4 ○5 ○6 ○7 Strongly agree

10. The feedback mechanism on Xianyu means that buyers and sellers can give evaluations to each other after the transaction is completed. I think [Scale question]

    The mechanism provides accurate information about a sellers' reputation.
    Strongly disagree ○1 ○2 ○3 ○4 ○5 ○6 ○7 Strongly agree

The mechanism has access to a wealth of useful information about the sellers' transaction history.
Strongly disagree ○1 ○2 ○3 ○4 ○5 ○6 ○7 Strongly agree
The mechanism would help me evaluate the sellers.
Strongly disagree ○1 ○2 ○3 ○4 ○5 ○6 ○7 Strongly agree

11. Regarding the Xianyu platform, I think [Scale question]

It is reliable.
Strongly disagree ○1 ○2 ○3 ○4 ○5 ○6 ○7 Strongly agree
It will keep its promises.
Strongly disagree ○1 ○2 ○3 ○4 ○5 ○6 ○7 Strongly agree
It is a trustworthy channel for me to transact.
Strongly disagree ○1 ○2 ○3 ○4 ○5 ○6 ○7 Strongly agree
The service offered by Xianyu meets my expectation.
Strongly disagree ○1 ○2 ○3 ○4 ○5 ○6 ○7 Strongly agree

12. As for the sellers on Xianyu, I think [Scale question]

They are in general trustworthy.
Strongly disagree ○1 ○2 ○3 ○4 ○5 ○6 ○7 Strongly agree
They are in general reliable.
Strongly disagree ○1 ○2 ○3 ○4 ○5 ○6 ○7 Strongly agree
They are in general honest.
Strongly disagree ○1 ○2 ○3 ○4 ○5 ○6 ○7 Strongly agree

13. Please make a judgment based on your willingness to trade in Xianyu. [Scale question]

I would consider transacting at Xianyu.
Strongly disagree ○1 ○2 ○3 ○4 ○5 ○6 ○7 Strongly agree
It is likely that I actually transact in Xianyu in the near future.
Strongly disagree ○1 ○2 ○3 ○4 ○5 ○6 ○7 Strongly agree
Given the opportunity, I intend to transact in Xianyu.
Strongly disagree ○1 ○2 ○3 ○4 ○5 ○6 ○7 Strongly agree

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
