# Peer review of "Impact of Community-Based Governance Mechanisms on Transaction Intention on a Second-Hand Trading Platform"

_jtaer, doi:10.3390/jtaer18010035_

Round 1

Reviewer 1 Report

Thank you for possibility to read this interesting paper. However, there are still some major issues to be done

1.) The overall structure needs to be significantly improved. The structure within the sections need also to be improved.
E.g. for the Introduction: In the introduction section, you need 4 paragraphs
- What do we already know?
- What do we not Know so far? --> derive the research gap in a clear an comprehensive form.
- How can we address this gap?
- Why is it important?

E.g.: for the Discussion: Why is a sub-chapter "main findings" ? Since you provide a quantitative study - you get results ! (Findings is the term for qualitative studies) and those "findings" are not to be found in the discussion section.

2.) Relatedly, he paper needs an in-depth discussion which connects this study to previous theories. In this format, the contributions are in much to high level. What is your contribution to the academic literature?

3.) The aim of the paper is presented in a bit confusing way.. The positioning of the paper and also the research gap remains fuzzy

4.) I would also like to see your definition about "governcence" - for instance Oliver Williamson (2010)?

Williamson, O. E., 2010. Transaction Cost Economics: The Natural Progression. Am. Econ. Rev. 100 (3), 673-690.

5.) Finally, the section 6 (discussion and conclusions) can be improved when the positioning of the paper and the research gap are clarified.

6.) Please, provide and align the definition for "governance mechanisms" - Your description is quite fuzzy.

7.) The language needs to be checked - it was mostly good but in some places the sentences were a bit difficult.

Author Response

Response to Reviewer 1 Comments

Point 1: The overall structure needs to be significantly improved. The structure within the sections need also to be improved.
E.g. for the Introduction: In the introduction section, you need 4 paragraphs

- What do we already know?

- What do we not Know so far? --> derive the research gap in a clear an comprehensive form.
- How can we address this gap?

- Why is it important?

E.g.: for the Discussion: Why is a sub-chapter "main findings" ? Since you provide a quantitative study - you get results ! (Findings is the term for qualitative studies) and those "findings" are not to be found in the discussion section.

Response 1: Thank you very much for detailed suggestions. We have reconstructed Introduction and Discussion. According to your suggestion, we decided to divide the Introduction into five paragraphs. The first paragraph explains the development of second-hand trading platform; The second paragraph describes the current literature, points out what we already know; The third paragraph points out the research gap which we don't know so far; The fourth paragraph explains our research questions and how we are going to do it; The fifth paragraph explains the implications of our research. We have already mentioned the four points you suggested which can be found in the section “Introduction”.

In addition, we also change the subchapter “Main Findings” into “Discussion” and rediscuss the contribution of this study which can be found in Line 536.

We apply the “Track Changes” function to mark those revisions. We hope the revised version is acceptable.

Point 2: Relatedly, he paper needs an in-depth discussion which connects this study to previous theories. In this format, the contributions are in much to high level. What is your contribution to the academic literature?

Response 2: Thank you very much for your suggestions. This study confirms that community-based governance on second-hand e-commerce platform can enhance transaction intention by enhancing two types of trust on the platform from three dimensions of relationship norms, conflict resolution and mutual dependence. Compared with previous studies, this study explores community-based governance from multiple dimensions and confirms the mediating role of trust in platform and trust in seller. In addition, the differences between consumers and prosumers are compared, and the boundary conditions of community-based governance are explored, contributing to the literature in the field of platform governance and online community.

Similarly, we have revised our discussion section to relate our conclusions to previous research in the line 536 to line 571 so that readers can be clear about our contribution.

“First of all, we did not study community-based governance from a social perspective like previous studies [7,24,29]. Based on the meaning of relationship governance [41], we divide community governance into three dimensions: relationship norms, conflict res-olution, and mutual dependence, and discuss the impact of three specific mechanisms on trust on second-hand trading platforms. This study distinguishes between two types of buyer trust on the platform. Previous studies have shown the difference between the two types of trust: trust in platforms is based on institutional guarantees, and trust in sellers is based on interpersonal interaction [44,45]. Through the research, we find that all the community-based governance mechanisms can improve transaction intention through two types of trust, in which the interest group has a higher impact on seller trust than platform trust, feedback mechanism and dispute resolution mechanism have little dif-ference on the two types of trust. This reflects that the community-based governance mechanisms can promote the enhancement of transaction intention on the platform from the two aspects of institution construction and interpersonal enhancement, but the impact of specific governance mechanisms is different. Compared with previous studies, this study confirms the role of community in enhancing platform governance capabilities from three dimensions of relationship norms, conflict resolution and interdependence.

In addition, this research explores the impact of community-based governance mechanisms on different types of buyers. In the second-hand trading platform, there is a new type of buyer, namely the prosumer, who participates in production activities and consumption activities. Previous studies have examined the participation motivation of prosumers [97] and the value that prosumers can provide to the platform[98], but have not considered the impact of platform governance on new production relationships. This study compares the differences between consumers and prosumers through multi-group analyses. The results show that the dispute resolution mechanism has a different impact on the two, and the dispute resolution mechanism has a significantly higher impact on consumers' trust in sellers and platforms than prosumers. This shows that the change of user identity will affect the results of community-based governance. When users par-ticipate in platform activities with multiple identities, some community-based governance will have a greater impact on users.”

Point 3: The aim of the paper is presented in a bit confusing way. The positioning of the paper and also the research gap remains fuzzy.

Response 3: Thank you for your valuable reviews. As mentioned in Response 1, based on your suggestions, we have clearly pointed out the research gap and our research positioning in line 66 to line 80.

"Previous studies took the community as a whole and explored the role of social in-teraction in the community on the e-commerce platform. However, it is necessary to establish and maintain the relationship between users through many specific governance mechanisms (such as interest groups, feedback evaluation, dispute resolution, etc.) from different dimensions. From the perspective of governance, the current research is not clear how different dimensions of community-based governance on the platform affect users. In addition, a few literatures have discussed the differences between users with different levels of community participation [13], but consumers participate in second-hand transactions in various formsincluding traditional consumers (who only participate in purchases) and prosumers (who participate in purchases and sales). It is not clear how the impact of community-based governance on these two groups of consumers differently. So, this study is guided by the following two research questions: How do different dimensions of community-based governance affect consumers’ transaction intentions? What is the difference between the impact of community-based governance on consumers and prosumers? "

Point 4: I would also like to see your definition about "governcence" - for instance Oliver Williamson (2010)?

Williamson, O. E., 2010. Transaction Cost Economics: The Natural Progression. Am. Econ. Rev. 100 (3), 673-690.

Response 4: Thank you very much for the paper you recommended to us. Through reading this paper, we have a deeper understanding of governance. From the perspective of transaction costs, governance is the means by which to infuse order, thereby to mitigate conflict and realize mutual gain. We thought the definition fit our research and added it which can be found in the line 213 and line 214.

Point 5: Finally, the section 6 (discussion and conclusions) can be improved when the positioning of the paper and the research gap are clarified.

Response 5: Thanks again for your suggestion. Based on your suggsetions, the section 6 of this paper is modified to clarify the positioning and research gap of the paper. In addtion, according to the suggestions of point 1 and 2, we changed the sub-chapter “Main Findings” into “Discussion”, explained the conclusion of this paper, and reviewed the previous papers to explain the contribution of this research to the previous research. However, due to the repetition of theoretical significance, we remove the subchapter “Theoretical Implications”.

Point 6: Please, provide and align the definition for "governance mechanisms" - Your description is quite fuzzy.

Response 6: Thanks again for your suggestion. We are very sorry for our negligence that we did not define the term "governance mechanisms".

As above we redefined governance as a mean to inject order into the platform and it mention a comprehensive mean. We refer to Ceccagnoli‘ descriptions and believe the governance mechanisms mentioned in the paper refer to the specific services and policies that platform owners provide to platform users which can be found in the line 214 to line 217.

Point 7: The language needs to be checked - it was mostly good but in some places the sentences were a bit difficult.

Response 7: Thanks again for your suggestion. We have re-read the paper and made revisions in the lines 22, 37, 282, 333, 367, 383, 439, 514, which we marked using the “Track Changes” function.

That’s all our changes and explanations. We have tried our best to improve the manuscript and we hope to meet with your approval. Special thanks to you. Furthermore, there are minor changes in the resubmitted manuscript about the grammar and modification of sentences. These changes will not influence the content and framework of the paper. We appreciate the reviewers’ warm and earnest work, and hope that the correction will meet with approval.

Once again, thank you very much for your comments and suggestions.

Reviewer 2 Report

Hello! I read your article, it's interesting but I think the article needs to be improved.

1. In the introduction, you need to add the relevance of this topic. It seems to me that today the popularity of such sites is declining, they have a lot of competition from social networks and streaming sites. Justify the relevance of your research.

2. The purpose of the article is not clearly defined.

3. Most of the links are outdated. In the introduction and in the literature review, the references needs to be reviewed and updated. Information technology has made a big step forward over the past 20 years, so the literature of the early 2000s is not always up to date.

4. In my opinion, you need to add a questionnaire in the appendix to the article. This article will only benefit from this.

5. It may be added in the Conclusion or in the Limitation that the findings are obtained for users or from China (if only they were interviewed). It remains unclear whether these conclusions can be extended to any users.

Author Response

Response to Reviewer 2 Comments

Point 1: In the introduction, you need to add the relevance of this topic. It seems to me that today the popularity of such sites is declining, they have a lot of competition from social networks and streaming sites. Justify the relevance of your research.

Response 1: Thank you for the insightful comments. Based on popular of social networks and streaming media sites, the improvement made by e-commerce platform is to build community on the platform and integrate community with e-commerce to increase the appeal to users. This study is based on this phenomenon. In recent years, some literature also has talked about the topic. For example:

  1. Luo, N.; Wang, Y.; Zhang, M.; Niu, T.; Tu, J. Integrating Community and E-Commerce to Build a Trusted Online Second-Hand Platform: Based on the Perspective of Social Capital. Technological Forecasting and Social Change 2020, 153, 119913, doi:10.1016/j.techfore.2020.119913.
  2. Molinillo, S.; Anaya-Sanchez, R.; Liebana-Cabanillas, F. Analyzing the Effect of Social Support and Community Factors on Customer Engagement and Its Impact on Loyalty Behaviors toward Social Commerce Websites. Comput. Hum. Behav. 2020, 108, 105980, doi:10.1016/j.chb.2019.04.004.
  3. Wang, J.; Cai, S.; Xie, Q.; Chen, L. The Influence of Community Engagement on Seller Opportunistic Behaviors in E-Commerce Platform. Electron. Commer. Res. 2021, doi:10.1007/s10660-021-09469-w.
  4. Fan, W.; Shao, B.; Dong, X. Effect of E-Service Quality on Customer Engagement Behavior in Community e-Commerce. Frontiers in Psychology 2022, 13, 965998, doi:10.3389/fpsyg.2022.965998.
  5. Myers, M.D.; Yang, X. Mothers’ Continuance Usage Intention Of A Pregnancy And Parenting Community E-Commerce Platform: Platform Gratifications And Mother Characteristics. Journal of Electronic Commerce Research 2020, 21, 277–293.

Point 2: The purpose of the article is not clearly defined.

Response 2: Thank you for your valuable reviews. Based on your suggestions, we have clearly pointed out the research gap and our research positioning in line 66 to line 80.

"Previous studies took the community as a whole and explored the role of social in-teraction in the community on the e-commerce platform. However, it is necessary to establish and maintain the relationship between users through many specific governance mechanisms (such as interest groups, feedback evaluation, dispute resolution, etc.) from different dimensions. From the perspective of governance, the current research is not clear how different dimensions of community-based governance on the platform affect users. In addition, a few literatures have discussed the differences between users with different levels of community participation[13], but consumers participate in second-hand transactions in various formsincluding traditional consumers (who only participate in purchases) and prosumers (who participate in purchases and sales). It is not clear how the impact of community-based governance on these two groups of consumers differently. So, this study is guided by the following two research questions: How do different dimensions of community-based governance affect consumers’ transaction intentions? What is the dif-ference between the impact of community-based governance on consumers and prosumers? "

We apply the “Track Changes” function to mark those revisions. We hope the revised version is acceptable.

Point 3: I recommend adding a separate "discussion" section. You can link it to the current conclusions section. Most of the links are outdated. In the introduction and in the literature review, the references needs to be reviewed and updated. Information technology has made a big step forward over the past 20 years, so the literature of the early 2000s is not always up to date.

Response 3: Thank you for your valuable comments. The literature has updated the introduction and literature review, adding a total of 7 literatures in recent 10 years. As follows:

  1. Zhang, J.; Qi, S.; Bei, L. A Receiver Perspective on Knowledge Sharing Impact on Consumer–Brand Relationship in Virtual Communities. Frontiers in Psychology 2021, 12, 685959, doi:10.3389/fpsyg.2021.685959.
  2. Goraya, M.; Zhu, J.; Shareef, M.; Imran, M.; Malik, A.; Akram, M. An Investigation of the Drivers of Social Commerce and E-Word-of-Mouth Intentions: Elucidating the Role of Social Commerce in E-Business. Electronic Markets 2019, 31, doi:10.1007/s12525-019-00347-w.
  3. Wu, J.; Huang, L.; Zhao, J.L.; Hua, Z. The Deeper, the Better? Effect of Online Brand Community Activity on Customer Purchase Frequency. Information & Management 2015, 52, 813–823, doi:10.1016/j.im.2015.06.001.
  4. Fan, W.; Shao, B.; Dong, X. Effect of E-Service Quality on Customer Engagement Behavior in Community e-Commerce. Frontiers in Psychology 2022, 13, 965998, doi:10.3389/fpsyg.2022.965998.
  5. Myers, M.D.; Yang, X. Mothers’ Continuance Usage Intention Of A Pregnancy And Parenting Community E-Commerce Platform: Platform Gratifications And Mother Characteristics. Journal of Electronic Commerce Research 2020, 21, 277–293.
  6. Storey, C.; Kocabasoglu Hillmer, C.; Roden, S.; de ruyter, ko Governing Embedded Partner Networks: Certification and Partner Communities in the IT Sector. International Journal of Operations & Production Management 2018, 38, doi:10.1108/IJOPM-12-2016-0708.
  7. Zhang, Y.; Li, J.; Tong, T.W. Platform Governance Matters: How Platform Gatekeeping Affects Knowledge Sharing among Complementors. Strateg. Manage. J. 2020, doi:10.1002/smj.3191.

Point 4: In my opinion, you need to add a questionnaire in the appendix to the article. This article will only benefit from this.

Response 4: Thank you for your suggestion. Based on your opinion, we have added a questionnaire at the appendix, hoping to get your approval.

Point 5: It may be added in the Conclusion or in the Limitation that the findings are obtained for users or from China (if only they were interviewed). It remains unclear whether these conclusions can be extended to any users.

Response 5: Thank you for your valuable suggestion. Based on the suggestion, we add the restrictions which can be found in the line 657 to the line 659.

The samples in this study are from China. It is unclear whether these conclusions can be generalized to any users. In the future, samples from all over the world can be collected on this issue to facilitate the generalization of conclusions.

That’s all our changes and explanations. We have tried our best to improve the manuscript and we hope to meet with your approval. Special thanks to you. Furthermore, there are minor changes in the resubmitted manuscript about the grammar and modification of sentences. These changes will not influence the content and framework of the paper. We appreciate the reviewers’ warm and earnest work, and hope that the correction will meet with approval.

Once again, thank you very much for your comments and suggestions.

Round 2

Reviewer 1 Report

Thank you - The paper can be considered accaptable.

Reviewer 2 Report

I think you did a good job. Thanks